# OpenReview forum: "Uncovering Model Vulnerabilities With Multi-Turn Red Teaming"
_ICLR.cc/2025/Conference — Submitted to ICLR 2025_

### Official Review · Reviewer_cegR · 2024-10-30

**Soundness:** 3
**Presentation:** 3
**Contribution:** 3
**Rating:** 8
**Confidence:** 4

**Summary:**

This paper looks at red teaming via multi-turn human interaction and releases an accompanying dataset of the jailbreaks. This is a useful study: the vast majority of the research effort and available data is focused on single-turn attacks, with only a few exceptions (e.g. Anthropic's hh-rlhf dataset having harmful multi turn conversations). Overall, the paper is clear and well written, and supplies useful well curated data for future work.

**Strengths:**

- Human driven red teaming data, particularly multi-turn due to its scarcity, is always useful. There are many synthetically generated datasets, but high quality human driven attacks are a valuable resource. In particular, each prompt here had a high degree of manual curation (in comparison to datasets like hackaprompt which contain many low quality samples).

- The analysis across a few defenses is useful, and does highlight that even SOTA defenses on highly studied benchmark harmful questions can be broken in around 15min on a large subset of questions (e.g. often quicker than algorithmic attack runtime).

**Weaknesses:**

- It would have been useful to see results which have human single turn jailbreak attempts in Figure 3. At the moment, the "Human" attacks have two variables changed compared to the others: the attack source (e.g. handcrafted), and additionally have multi-turn capabilities. Hence it makes it challenging to disambiguate if the difference in performance is due to the multi-turn aspect, or if humans given enough time remain better than automated based methods at creating jailbreaks.
- I am unclear as to why results against Cygent defense could not be carried out in the same setup as the original paper: the Llama model is open source, and the defense has a published paper. It would have enabled stronger reproducibility and clearer interpretation of results.
- Releasing the non-successful jailbreak attempts as well can be beneficial as it is still a useful resource, for example as training/fine-tuning data or to carry out further analysis.
- Although different styles of harmfulness were investigated: both "regular" harmbench style questions, but also WMDP-Bio for different attack objectives and domain performance from looking into the supplementary material it seems like just the harmbench data was released. Given the dataset is the core contribution of the paper it would have been useful to include the other domain data.

**Questions:**

Are there statistics on how many red-team members there are/distributions on number of samples provided per red-team member?

---

> ### Author Response · Authors · 2024-11-16
> **Author Rebuttal**
>
> Hi, thank you for the kind review and pointing out some areas for clarity within the paper. We are glad to hear that you find the quality of the dataset useful! We hope we have clarified your concerns below:
>
> **Weaknesses**
> 1. _It would have been useful to see results which have human single turn jailbreak attempts in Figure 3. At the moment, the "Human" attacks have two variables changed compared to the others: the attack source (e.g. handcrafted), and additionally have multi-turn capabilities. Hence it makes it challenging to disambiguate if the difference in performance is due to the multi-turn aspect, or if humans given enough time remain better than automated based methods at creating jailbreaks._
> - We acknowledge a limitation that hinders this analysis. The instructions we gave to the red teamers did not limit the number of turns. Since we optimized under the multi-turn threat model, our human-single turn jailbreaks were too skewed to disambiguate the cause. We found 91.6% of human jailbreaks were multi-turn.
>
> 2. _I am unclear as to why results against Cygent defense could not be carried out in the same setup as the original paper: the Llama model is open source, and the defense has a published paper. It would have enabled stronger reproducibility and clearer interpretation of results._
> - CYGNET is not open source (in particular, the system prompt is not public), and while their paper is published we cannot reproduce critical details. We instead report the attack results from the CYGNET authors given we are using the same HarmBench framework for consistency, and we use their API for our human red teaming.
>
> 3. _Releasing the non-successful jailbreak attempts as well can be beneficial as it is still a useful resource, for example as training/fine-tuning data or to carry out further analysis._
> - We considered this and did look through the non-successful jailbreaks. For context, red teamers are generally free to start and restart conversations, frolicking over many attempts in 30 minutes. As you can imagine, we found the unsuccessful jailbreak attempts from this process were particularly unhelpful and noisy, and did not believe they would be a good contribution to the community. We chose to release the highest quality data, which happens to be the successful jailbreaks.
>
> 4. _Although different styles of harmfulness were investigated: both "regular" harmbench style questions, but also WMDP-Bio for different attack objectives and domain performance from looking into the supplementary material it seems like just the harmbench data was released. Given the dataset is the core contribution of the paper it would have been useful to include the other domain data._
> - Legal reasons: Thank you for looking through the supplementary materials! This is a deliberate choice. At the advice of our legal counsel, we cannot release any data from the WMDP-Bio jailbreaks in accordance with ITAR and EAR (see Section 7 - Ethics). We omitted certain jailbreak prompts from the HarmBench set for the same reason.
>
>
> **Questions**
> 1. _Are there statistics on how many red-team members there are/distributions on number of samples provided per red-team member?_
> - We had 13 red teamers in total. For each defense, we included at least 6 red teamers to capture a diversity of experiences. At the same time, to reduce variability in aggregate due to human skillsets (and to be fair to workload), each red teamer attempted approximately the same number of prompts.

---

> > ### Author Response · Authors · 2024-11-24
> > **Conclusion of Discussion Period**
> >
> > Thank you for your thoughtful review and for recognizing the value of our multi-turn human red teaming dataset, particularly noting its high degree of manual curation! I hope we've adequately addressed your questions about methodology by explaining our intentional focus on multi-turn attacks (with 91.6% of successful jailbreaks being multi-turn), clarifying the CYGNET reproducibility constraints, and detailing our curation process which prioritized releasing high-quality successful jailbreaks while adhering to legal requirements regarding sensitive data. As the discussion period concludes, we appreciate your support of our work and welcome any additional questions about our methodology or dataset.

---

> > > ### Comment · Reviewer_cegR · 2024-11-26
> > >
> > > Thank you for the clarifications. With the current score of 8/accept reflects my assessment of the paper; I do not feel the concerns I raised/have been addressed were significant enough to increase the score further.

---

### Official Review · Reviewer_V88E · 2024-11-02

**Soundness:** 3
**Presentation:** 3
**Contribution:** 2
**Rating:** 6
**Confidence:** 4

**Summary:**

This paper exposes the significant safety vulnerabilities of Large Language Models (LLMs) in multi-turn dialogue by conducting human red-teaming. By exploiting a diverse of human tactics, human red-teamers achieve a 70% attack success rate on Harmbench against strong defense baselines. This paper finally compiles these results into the Multi-Turn Human Jailbreaks (MHJ) dataset.

**Strengths:**

+ **Valuable Findings, Including an Open-Source Jailbreak Dataset**: This paper provides a valuable contribution by conducting the first multi-turn human red-teaming experiments, revealing significant safety vulnerabilities that bypass current state-of-the-art defense methods. Given that existing safety alignment techniques primarily target single-turn jailbreaks, these findings effectively motivate the community to develop more robust defenses capable of countering multi-turn attacks.

**Weaknesses:**

- **Lack of Strong Automated Attack Baselines**: Some black-box attacks, such as CodeAttack [1] and PAP [2], are not included in the experiments, despite utilizing similar “Request Framing” tactics as those in this paper. Including these baselines in future comparisons would offer a clearer perspective on the relative effectiveness of this paper's approach.

- **Unclear Evaluation Metric for Model Unlearning**: This paper proposes manual review to assess attack outcomes in model unlearning experiments, yet does not clearly define what constitutes a successful attack. For example, does success mean the model output includes content from the forget set? A more detailed description of the evaluation metric for model unlearning experiments would improve clarity.

1. *CodeAttack: Revealing Safety Generalization Challenges of Large Language Models via Code Completion.*
2. *How Johnny Can Persuade LLMs to Jailbreak: Rethinking Persuasion to Challenge AI Safety by Humanizing LLMs.*

**Questions:**

- **Counter-Intuitive Experimental Results**: The effectiveness of the “Direct Request” tactic used by human red-teamers, as shown in Figure 8, is unexpected. In this paper, “Direct Request” refers to directly asking the LLM to produce harmful content, which should logically not be highly effective against safety defenses. An analysis of how various tactics influence the attack success rate would be beneficial, as it could offer insights into the specific failure modes of these models.

- **Lack of In-Depth Analysis on the High Effectiveness of Human Multi-Turn Jailbreaks**: Many of the tactics employed by human red-teamers, such as “Injection,” “Obfuscation,” “Output Format,” and “Request Framing,” are also used by automated attacks. Are there particular factors that enable human-led jailbreaks to outperform automated attacks using similar tactics? Identifying these factors could provide valuable insights into the unique strengths of human-driven attacks.

I would consider raising my score if the authors address these concerns and questions.

---

> ### Author Response · Authors · 2024-11-16
> **Author Rebuttal (Weaknesses)**
>
> Thank you for your clear insights within the scope of our paper! We are happy to hear that you find our contributions valuable to the community, including our open sourced dataset. We addressed your concerns below, and we are happy to continue discussing our work further in this period.
>
> **Weaknesses**
> 1. _Lack of Strong Automated Attack Baselines: Some black-box attacks, such as CodeAttack [1] and PAP [2], are not included in the experiments, despite utilizing similar “Request Framing” tactics as those in this paper. Including these baselines in future comparisons would offer a clearer perspective on the relative effectiveness of this paper's approach._
> - We considered PAP, but it is a weaker attack than the PAIR and GCG attacks we already implement – Table 6 in the Appendix of HarmBench [3] demonstrates that PAIR and GCG outperform PAP-top5 on all 20 open-source, and PAIR outperforms PAP on all 8 closed-source models tested. We agree it will help completeness, but we do not think it will add value in terms of showcasing the effectiveness of our attack.
> - CodeAttack is an old attack from more than 2 years ago and is not implemented within the HarmBench framework, the largest standardized collection of adversarial attacks. We do not believe that it would be strong compared to the more modern attacks we test.
> 2. _Unclear Evaluation Metric for Model Unlearning: This paper proposes manual review to assess attack outcomes in model unlearning experiments, yet does not clearly define what constitutes a successful attack. For example, does success mean the model output includes content from the forget set? A more detailed description of the evaluation metric for model unlearning experiments would improve clarity._
> - Thanks for catching this missing detail. Since WMDP-bio test split is a labeled multiple-choice dataset, we simply use the ground truth answer corresponding to each prompt to verify the attack. This is not an exact string match, but rather that the answer from a jailbroken model output semantically contains all the information in the label, as judged by a human reviewer. We will add this detail in our OpenReview submission.
>
> [3] Mazeika, M., Phan, L., Yin, X., Zou, A., Wang, Z., Mu, N., ... & Hendrycks, D. (2024). Harmbench: A standardized evaluation framework for automated red teaming and robust refusal. arXiv preprint arXiv:2402.04249.

---

> > ### Author Response · Authors · 2024-11-16
> > **Author Rebuttal (Questions)**
> >
> > **Questions**
> > 1. _Counter-Intuitive Experimental Results: The effectiveness of the “Direct Request” tactic used by human red-teamers, as shown in Figure 8, is unexpected. In this paper, “Direct Request” refers to directly asking the LLM to produce harmful content, which should logically not be highly effective against safety defenses. An analysis of how various tactics influence the attack success rate would be beneficial, as it could offer insights into the specific failure modes of these models._
> > - We are not sure about the exact reason, which is why we wanted to release this paper to the wider community and highlight the vulnerability. Our leading hypothesis comes from the following: Direct Request over a multi-turn context is different from directly asking in a single turn. For example, instead of asking the model to directly generate harmful code, the red teamer can provide some code and incrementally ask the model to modify it to become increasingly harmful, while being direct in intention. It’s more helpful to think of the tactics as clusters instead of hard strategies since our human red teamers are not constrained like automated attacks and they self-label their own tactic. As a result, direct requests can include a wide variety of strategies that use the full threat of multiple turns, as we release in our dataset.
> > - We also note Direct request is often the first tactic that all our red teamers try. Once they find a tactic that jailbreaks the model, they do not try other tactics (it would be infeasible to exhaustively try all tactics within a time limit). Hence, the tactic we report is often the first tactic that works, not that it is the only tactic that works.
> > 2. _Lack of In-Depth Analysis on the High Effectiveness of Human Multi-Turn Jailbreaks: Many of the tactics employed by human red-teamers, such as “Injection,” “Obfuscation,” “Output Format,” and “Request Framing,” are also used by automated attacks. Are there particular factors that enable human-led jailbreaks to outperform automated attacks using similar tactics? Identifying these factors could provide valuable insights into the unique strengths of human-driven attacks._
> > - Some automated attacks are analogous to these tactics. These methods are usually very constrained, they do not try again with a different method on failure or search over multiple turns. Our setup involves a human trying all sorts of tactics within a time limit, expressing high creativity even within a tactic. The key strength is the adaptability of the search space in a human jailbreak, instead of only optimizing for the execution of a single tactic as automated jailbreaks do. Hence, we do not believe the success of MHJ is because humans execute similar tactics better.
> > - We approximated a wider search space for automated attacks with the ensemble attack, however, it still reveals a large gap between human and automated attacks. We leave the exploration of how to select and adapt diverse tactics automatically to future work.

---

> > > ### Author Response · Authors · 2024-11-24
> > > **Conclusion of Discussion Period**
> > >
> > > Thank you for your informative review, and for highlighting our paper's valuable contribution in revealing multi-turn safety vulnerabilities and providing an open-source jailbreak dataset! I hope we've addressed your concerns by explaining our baseline selection, including noting that PAIR and GCG outperform other methods in comprehensive testing, clarifying our evaluation metrics for model unlearning, and providing insight into the effectiveness of Direct Request tactics. As the discussion period concludes, we would greatly appreciate it if you could consider raising the score to reflect how our responses have addressed your concerns!

---

> > > > ### Comment · Reviewer_V88E · 2024-12-02
> > > > **Thanks for your response**
> > > >
> > > > Thank you for your response. My concerns have been partially addressed, although I still have some doubts about whether direct requests are truly an effective human tactic. I suggest the authors check for possible noise in human labeling. I have decided to raise my score to 6.

---

> > > > > ### Author Response · Authors · 2024-12-02
> > > > > **Thank you for your review**
> > > > >
> > > > > We thank the reviewer for raising their score and for their feedback. We will investigate the possibility of human noise in our tactic labelling.

---

### Official Review · Reviewer_qGEv · 2024-11-04

**Soundness:** 3
**Presentation:** 3
**Contribution:** 3
**Rating:** 6
**Confidence:** 3

**Summary:**

This paper focuses on the safety vulnerabilities of LLMs in multi-turn conversations. The authors designed a comprehensive human red teaming pipeline, including "Attempt Jailbreak" and "Validate Jailbreak," and found that LLMs exhibit more safety risks in multi-turn dialogue conversations compared to single-turn automated attacks.

**Strengths:**

1. Most previous papers have focused on LLM jailbreaking in single-turn conversations, while this work emphasizes multi-turn conversations, presenting a new scenario.

2. The red-teaming results show that multi-turn human jailbreaks outperform current automated attacks, which exposes more safety vulnerabilities in LLMs.

3. The authors have also released their Multi-turn Human Jailbreaks dataset, which can support further research on multi-turn jailbreaks.

**Weaknesses:**

1. The human red-teaming was conducted only on Llama; further evaluations on other LLMs would more comprehensively illustrate the safety vulnerabilities in multi-turn conversations. Additionally, the human red-teaming data collected on Llama could potentially be used to examine safety issues in other LLMs under multi-turn scenarios.

2. A more detailed analysis of the effectiveness of different tactics could provide deeper insights into multi-turn jailbreaks.

**Questions:**

NA

---

> ### Author Response · Authors · 2024-11-16
> **Author Rebuttal**
>
> Hello, thank you for your review! We are glad that you find our paper to be comprehensive, dataset useful, and that this work reveals important safety vulnerabilities. We’ve addressed some of your concerns, please let us know if there’s anything else we can clarify.  We look forward to a discussion with you.
>
> **Weaknesses**
> 1. _The human red-teaming was conducted only on Llama; further evaluations on other LLMs would more comprehensively illustrate the safety vulnerabilities in multi-turn conversations. Additionally, the human red-teaming data collected on Llama could potentially be used to examine safety issues in other LLMs under multi-turn scenarios._
> - Our goal was to show if multi-turn attacks lead to a high ASR against the strongest defenses, which in prior work are generally implemented on Llama3-8b. We do not need to experiment on models we know are already weak to single-turn attacks (Appendix Table 6, HarmBench: https://arxiv.org/abs/2402.04249). We do not believe that evaluating MHJ on other LLMs would improve the contribution towards our goal, but we recognize that it would be scientifically interesting to try to find more human attack patterns on different models if we had additional resources.
> 2. _A more detailed analysis of the effectiveness of different tactics could provide deeper insights into multi-turn jailbreaks._
> - We acknowledge this as a limitation. The analysis is brief for two reasons: (1) We do not exhaustively try all possible tactics (infeasible), instead we have red teamers freely try tactics until one works and stop when a jailbreak is successful. (2) The tactics are not rigid, but rather self-reported clusters by the red teamers. Hence, the diverse nature of the attack, even within a tactic, makes it very difficult to generalize patterns. Based on our interactions with the red teamers, we believe the fundamental tactic is adaptability of human red teamers, rather than the specific tactic they first used.
> Instead of an unclear summarization, we decided to open-source the full  with the hope to inspire others in the community to find patterns and build stronger attacks and defenses._
>
> **Questions**
> None from reviewer.

---

> > ### Author Response · Authors · 2024-11-24
> > **Conclusion of Discussion Period**
> >
> > Thanks once again for your thoughtful review! We appreciate that you recognize the importance and novelty of our work through demonstrating the effectiveness of the multi-turn jailbreak paradigm and release of our open-source dataset. I hope we've addressed all of your concerns, including clarifying how our analysis of tactics was intentionally focused on demonstrating human adaptability rather than rigid categorization. As the discussion period concludes, we would greatly appreciate it if you could consider raising the score to reflect how our responses have addressed your concerns!

---

> > ### Comment · Reviewer_qGEv · 2024-11-24
> >
> > Thank you for the authors' response. Since I have already given a positive score, I will keep it.

---

### Official Review · Reviewer_EdXj · 2024-11-04

**Soundness:** 3
**Presentation:** 2
**Contribution:** 1
**Rating:** 3
**Confidence:** 4

**Summary:**

The paper highlights the limitations of current defenses in large language models (LLMs) against multi-turn adversarial attacks. Unlike single-turn automated attacks, multi-turn human-driven jailbreaks show a much higher success rate, bypassing existing defenses by strategically interacting with models over multiple conversation turns. The authors conducted tests with human red teamers who revealed that many LLMs, despite advanced safeguards, remain vulnerable in realistic settings. The study also introduces the "Multi-Turn Human Jailbreaks" (MHJ) dataset, containing thousands of successful jailbreak examples to support the development of more robust defenses that address multi-turn interaction vulnerabilities.

**Strengths:**

1. The main strength of the paper is evaluating the LLM jailbreak robustness from a different perspective and in a like-life setting. A realistic threat model can always give a better idea about the robustness.

2. I appreciate the authors for publishing the dataset of 2912 prompts. I believe this can help the researchers and community.

**Weaknesses:**

1. Only the *Llama3-8b-Instruct* model was used for evaluation. Other models, especially the stronger ones should have been included.

2. The chosen baselines and the multi-turn-human-jailbreak approach are different from each other in many ways, such as timing, budget, etc. Seems like the MHJ attack is a soft-constrained version of attacks.

3. Authors admit that *"the skill and experience of individual red teamers may vary"*. In that case, how can researchers compare their defense against human-based attacks in the common ground? What would be the baseline of a red-teamer's skill and experience? This is one of the reasons for using automated attacks for evaluation to this date.

4. Other automated attacks, such as [1, 2] could be included in the evaluation.

5. There was no discussion on plausible defenses for such multi-turn-human-based attack.


References:


[1] Fast Adversarial Attacks on Language Models In One GPU Minute

[2] Great, Now Write an Article About That: The Crescendo Multi-Turn LLM Jailbreak Attack

**Questions:**

1. Why the evaluation was done only on one model? Why stronger models like ChatGPT was not included?

2. How were other automated attacks implemented for the WMDP-bio questions?

3. Is it possible to mention the monetary cost of this human-based attack?

---

> ### Author Response · Authors · 2024-11-16
> **Weaknesses (Part 1)**
>
> Hi, thank you for your review! We wanted to clarify our choices. In short, our choice of attacks were aimed at being consistent with prior work, while choosing some of the strongest defenses we could find. We look forward to a fruitful discussion, please let us know if you have any additional questions.
>
> **Weaknesses**
> 1. _Only the Llama3-8b-Instruct model was used for evaluation. Other models, especially the stronger ones should have been included._
> - Consistency with prior work: The defenses we evaluated were all implemented on the 8b version of Llama. We made this choice to be consistent with prior baselines. In particular, CYGNET, which is closed source, only existed in the Llama-3-8b form at the time of our experiments.
> - Stronger models: We focus on defenses (models with enhanced guardrails), which when implemented on Llama-3-8b is already incredibly robust against single-turn attacks. Stronger capability should not be confused with robustness. Our task is more difficult, for example, CYGNET has a 0% ASR on all the automated attacks tested. We also detail in the question section below that larger models such as ChatGPT are not necessarily more robust, as prior work in the field discovered.
>
> 2. _The chosen baselines and the multi-turn-human-jailbreak approach are different from each other in many ways, such as timing, budget, etc. Seems like the MHJ attack is a soft-constrained version of attacks._
> - We acknowledge the difference: We do not claim that MHJ is analogous in time or computational budget. In fact, MHJ may be orders of magnitude more expensive. Our goal is to demonstrate a jailbreak that worked, where prior automated attacks would not scale even with more budget.
> - Automated attacks don’t scale: At least, it is unclear how to scale automated attacks. For example, we empirically found if GCG does not successfully jailbreak in the first 1000 steps, 10x more optimization steps will not give a jailbreak (imagine a loss curve that has converged and cannot go further down). Hence, we did not believe it was a meaningful use of resources to do budget-inflated experiments with automated attacks.
> - We are unsure to understand the meaning of “soft-constrained” attacks. Can you clarify this? We give our human red teamers limited access via a chat interface, but unlimited freedom in terms of attack strategy, rather than the highly constrained search spaces of automated attacks.
>
>
> 3. _Authors admit that “the skill and experience of individual red teamers may vary”. In that case, how can researchers compare their defense against human-based attacks in the common ground? What would be the baseline of a red-teamer’s skill and experience? This is one of the reasons for using automated attacks for evaluation to this date._
> - Fundamental limitation of human studies: Human variance is a limitation in any study, we cannot guarantee people are uniform. The text “the skill and experience of individual red teamers may vary” was exactly meant to remind this to readers and highlight that this study might suffer from these drawbacks by construction. In this context, human variance might reduce exact reproducibility of the experiments but it won’t invalidate the soundness of our results. We stress that in jailbreaking studies, one configuration (in this case the slice of the population) breaking the model is typically enough to consider the attack valid. Especially if the configuration, despite these limitations, achieves a 70% ASR on CYGNET, a model where prior automated attacks achieve 0%.
> - Reducing human variance: Our red teamers are all experienced employees with standardized training, who have red teamed many commercial frontier language models. Each defense is assigned 6 distinct red teamers in the pipeline, and each human is assigned to attempt jailbreaking approximately the same number of prompts to reduce the effect of human variance. We acknowledge that the proprietary nature of their training creates barriers to reproducing our work, hence we release the dataset of jailbreaks publicly to inspire more work in this area.

---

> > ### Author Response · Authors · 2024-11-16
> > **Weaknesses (Part 2)**
> >
> > 4. _Other automated attacks, such as [1, 2] could be included in the evaluation._
> > - Why not more attacks? Neither of these attacks have been implemented in the standardized HarmBench framework we use. We already compare against the 6 automated attacks with high ASR against the defenses we tested. We believe the message of our work is clear with the choice of our existing attacks. We are open to cite other existing attacks, if relevant to this work.
> > - Crescendo: We cited the Crescendo attack [2] as we believed it was highly relevant in proposing the multi-turn threat model, and would have liked to compare against it. Crescendo is not open source, hence we could not include it in our evaluation without knowing how to exactly reproduce the attack.
> >
> > 5. _There was no discussion on plausible defenses for such multi-turn-human-based attack._
> > - Objective: The goal of our work is to point out a false sense of robustness that single-turn defenses provide. Developing defenses against multi-turn attacks are out of the scope of our work as explained below, nevertheless this is a fair comment we address here.
> > - Defenses: In brief, we found the human attacks to be very crafty and unstructured (see our dataset in the supplementary materials), at the moment we cannot think of any method to defend against a pipeline of human red teamers! A couple ideas floated around such as rate limitation and flagging high risk accounts, but this is a different threat model and meta-defense focused, hence we did not include it in the discussion out of relevance.
> >
> > **Questions**
> > 1. _Why the evaluation was done only on one model? Why stronger models like ChatGPT was not included?_
> > - The defenses we use are much stronger than ChatGPT and amongst the strongest open-source defenses we could find at the time of writing the paper. ChatGPT would not be an interesting baseline because automated attacks are already very effective against them (see Table 6 in HarmBench: https://arxiv.org/abs/2402.04249)
> > 2. _How were other automated attacks implemented for the WMDP-bio questions?_
> > - There was no change in implementation, we use the standard HarmBench implementation (https://github.com/centerforaisafety/HarmBench) and simply use the WMDP-bio questions as the prompt.
> > 3. _Is it possible to mention the monetary cost of this human-based attack?_
> > - Unfortunately we cannot reveal the monetary cost, as the red teamers we use are from a commercial firm. We acknowledge this response is not satisfying but hope you can understand this limitation.

---

> > > ### Author Response · Authors · 2024-11-24
> > > **Conclusion of Discussion Period**
> > >
> > > We once again thank you for your thorough review, and particularly for recognizing the value of our realistic threat model evaluation and the public release of our 2,912-prompt dataset, which we believe will significantly benefit the research community. We hope we've addressed your concerns regarding model selection by explaining our focus on defense-enhanced models (where Llama-3-8b with CYGNET achieves 0% ASR against all tested automated attacks) and clarified the methodological rationale behind our human red-teaming approach, including our efforts to standardize the process through experienced testers and consistent prompt distribution. As the discussion period concludes, we would greatly appreciate it if you could consider raising the score to reflect how our responses have addressed your concerns!

---

> > > > ### Comment · Reviewer_EdXj · 2024-11-26
> > > > **Thanks to authors**
> > > >
> > > > I appreciate the authors' effort in this rebuttal. After the rebuttal, I have decided to keep my score unchanged.

---

> > > > > ### Author Response · Authors · 2024-11-26
> > > > > **Thanks to Reviewer EdXj**
> > > > >
> > > > > Thanks so much for your review and response! We believe our rebuttal has addressed your concerns in detail, and would appreciate it if you could be more specific about any further concerns. Could you please describe why your score remains unchanged, and what additional changes are necessary for a raised score? Thanks again for your help!

---

> > > > > > ### Author Response · Authors · 2024-12-02
> > > > > > **Conclusion of Discussion Period and Request For Feedback**
> > > > > >
> > > > > > We once again appreciate your feedback and recognition of our response! Given that our rebuttal has addressed each point in detail, it would again be appropriate if you could share any specific remaining concerns or suggestions for improvement that would help us strengthen the paper. As the other reviewers have raised their scores to 6, 6, and 8 following our responses, your specific guidance would be particularly valuable in understanding what additional changes would be necessary to address your concerns and warrant a score revision. Thanks again for your help!

---

### Author Response · Authors · 2024-11-16
**Overall Author Response**

Thank you to the reviewers for their detailed feedback! We’ve responded individually to every point of concern and question, in addition to providing the summary below.

We observe some disagreement between reviewers, with reviewer 4 giving the highest score after studying the supplemental dataset. This rebuttal aims to clarify the soundness and value of our contribution.

**Summary of reviewer agreed strengths and key contributions:**
- **Valuable results**: The paper shows multi-turn human red teaming easily breaks SOTA defenses (robust against single turn attacks). The strong performance of this realistic threat model highlights a need to develop better defenses.
- **High quality human dataset**: Our released dataset of human jailbreaks is an important contribution because of the high quality of manual curation. Reviewers agree this dataset will further support researchers in the field.

**We highlight two common concerns from reviewers 1, 2, and 3.** We note these concerns do not harm the findings of the paper, but we agree more experiments would enhance completeness. However, human experiments are incredibly costly and time-consuming, hence we were forced to make some design choices to adhere to work the community has already found valuable.

- **We should attack “stronger models”**:  We should not confuse capability with adversarial robustness. Prior work such as HarmBench (Appendix Table 6) [1] has shown that other models suggested, such as ChatGPT, are substantially less robust. We focus on the more difficult task of attacking defenses. The defenses we experiment with are already some of the most robust defenses against single-turn attacks at the time of experimentation. We use defenses on top of Llama3-8b to be consistent with prior work, it does not mean this model is less robust than larger models. For example, CYGNET has a 0% ASR on all other automated attacks as reported by the original authors, whereas we achieve a 70% ASR. Since we already break strong defenses, we do not see a need to add more less robust models to attack.
- **Implementing additional attacks**: The addition of more recent attacks would not substantially change the message of the paper: defenses which claim to be robust to single-turn attacks are not robust to multi-turn human attacks yet, as both a factor of the multi-turn objective and the unconstrained creativity of human red teamers. We chose to use a standard framework that has attacks implemented, in this case HarmBench [1]. There are many attacks which have come out recently not part of this standardized library, we simply selected 6 in HarmBench that were straightforward or did well on defenses.

**References**

[1] Mazeika, M., Phan, L., Yin, X., Zou, A., Wang, Z., Mu, N., ... & Hendrycks, D. (2024). Harmbench: A standardized evaluation framework for automated red teaming and robust refusal. arXiv preprint arXiv:2402.04249.

---

### Meta-Review · Area_Chair_dqCs · 2024-12-16

**Metareview:**

There have been several multi-turn jailbreaking attacks proposed in the past, such as Crescendo, mentioned by the authors. This reduces the novelty and impact of the proposed work. Moreover, the authors have not adequately compared their method with critical prior works like Crescendo to highlight key differences. While the authors attribute the lack of comparison to the unavailability of open-sourced resources, the code for Crescendo is publicly accessible. Additionally, there are other multi-turn jailbreaking attacks that should also be included for a fair comparison. The AC recommends that the authors conduct a more comprehensive evaluation, not only focusing on ASR but also considering factors like prompt diversity and the cost introduced by the proposed attack method.

**Additional Comments On Reviewer Discussion:**

The authors have addressed some of the reviewers' concerns. However, the primary issue remains the novelty of the multi-turn jailbreak attack. This paper is not the first work in this sub-area, and as such, the reviewers strongly recommend including more comprehensive comparisons. The authors' reasoning for not conducting additional experiments is not convincing.

---

### Decision · Program_Chairs · 2025-01-22

Reject